# In-Silico Monte Carlo Simulation Trials for Investigation of V_2_O_5_ Reinforcement Effect on Ternary Zinc Borate Glasses: Nuclear Radiation Shielding Dynamics

**DOI:** 10.3390/ma14051158

**Published:** 2021-03-01

**Authors:** Huseyin O. Tekin, Shams A. M. Issa, Gokhan Kilic, Hesham M. H. Zakaly, Mohamed M. Abuzaid, Nevzat Tarhan, Khatar Alshammari, Hj Ab Aziz Sidek, Khamirul A. Matori, Mohd H. M. Zaid

**Affiliations:** 1Medical Diagnostic Imaging Department, College of Health Sciences, University of Sharjah, Sharjah 27272, United Arab Emirates; tekin765@gmail.com (H.O.T.); mabdelfatah@sharjah.ac.ae (M.M.A.); 2Medical Radiation Research Center (USMERA), Uskudar University, 34672 Istanbul, Turkey; nevzat.tarhan@uskudar.edu.tr; 3Physics Department, Faculty of Science, Al-Azhar University, Assiut 71524, Egypt; shams_issa@yahoo.com; 4Physics Department, Faculty of Science, University of Tabuk, Tabuk 71451, Saudi Arabia; 5Department of Physics, Eskisehir Osmangazi University, 26040 Eskisehir, Turkey; gkilic@ogu.edu.tr; 6Institute of Physics and Technology, Ural Federal University, 620000 Ekaterinburg, Russia; 7NP Istanbul Brain Hospital, 34768 Istanbul, Turkey; 8Department of Physics, Faculty of Arts and Sciences, Northern Border University, Turaif 75311, Saudi Arabia; alshammarikhatar1@gmail.com; 9Faculty of Medical Sciences, Newcastle University, Newcastle NE2 4HH, UK; 10Department of Physics, University Putra Malaysia, Serdang 43400, Selangor, Malaysia; sidek@upm.edu.my (H.A.A.S.); khamirul@upm.edu.my (K.A.M.)

**Keywords:** vanadium pentoxide, Monte Carlo simulation, ternary zinc borate, radiation shielding dynamics

## Abstract

In the current study, promising glass composites based on vanadium pentoxide (V_2_O_5_)-doped zinc borate (ZnB) were investigated in terms of their nuclear-radiation-shielding dynamics. The mass and linear attenuation coefficient, half-value layer, mean free path, tenth-value layer, effective atomic number, exposure-buildup factor, and energy-absorption-buildup factor were deeply simulated by using MCNPX code, Phy-X PSD code, and WinXcom to study the validation of ZBV1, ZBV2, ZBV3, and ZBV4 based on (100−x)(0.6ZnO-0.4B_2_O_3_)(x)(V_2_O_5_) (x = 1, 2, 3, 4 mol%) samples against ionizing radiation. The results showed that attenuation competencies of the studied glasses slightly changed while increasing the V_2_O_5_ content from 1 mol% to 4 mol%. The domination of ZnO concentration in the composition compared to B_2_O_3_ makes ZnO substitution with V_2_O_5_ more dominant, leading to a decrease in density. Since density has a significant role in the attenuation of gamma rays, a negative effect was observed. It can be concluded that the aforementioned substitution can negatively affect the shielding competencies of studied glasses.

## 1. Introduction

Glasses, which are flexible materials in terms of ease of production and structural variations to production, have vital roles in technology and research [1]. Utilized as the primary raw materials, phosphorus pentoxide, boron oxide, and vanadium pentoxide are the most common compounds in the manufacture of glasses [2,3,4]. Of these chemicals, B_2_O_3_ is regarded as the strongest glass former [5,6,7]. Borate glasses in which B_2_O_3_ establishes the glass network are mostly used as optical materials because they have low melting points, high transmittance properties, and high thermal stability [8,9,10]. They are widely used in the processing of dielectric materials, and used in electronic materials. In addition to their use as dielectric components, the presence of transition metal ions in the borate glass network results in a semiconducting character [11,12,13,14,15]. Transition metals are used in glass science in different contexts because of their complex properties that emerge from several valence states [16]. Zinc oxide/borax glasses are being used in plasma screens and displays for high definition and quality. They are chosen due to their ability to quickly pass energy in dielectric layers and their high transparency [17]. With all of the qualities mentioned above, zinc borate glasses stand out among products for applications that are in doubt. Studies on vanadium have grown recently due to the material’s curious optical, electrical, and magnetic properties [18,19,20,21,22,23]. Though studies on transition-metal-doped glasses can frequently be found in the literature, there are still components, compositions, and properties that have not yet been studied. Despite the broad variety of papers researching transition-metal-doped glasses, many elements, formulations, and properties are not yet researched. On the other hand, analysis suggests that these glasses can be extremely protective against radiation [24,25,26,27,28,29,30,31,32,33]. The use of the glass material for radiation shielding is not limited to this kind of glass, but the frequency of experimental studies on the issue has been increasing daily as a hot topic. Any conventional materials such as lead and concrete used in nuclear fields do not possess superior material properties and have some features that can endanger health [34]. Although this report does not say unequivocally that these new shielding materials are an insufficient radiation-shielding material, some organizations such as IAEA have direct incentives for researchers to study new-generation shielding materials. In this case, glasses suit the profile of these kinds of uses. For this aim, the compositions of a group of glasses encoded as ZBV1, ZBV2, ZBV3, and ZBV4 [35] were tested and extensively researched for their capabilities in gamma-radiation attenuation.

In this study, gamma-attenuation properties of the ZnO-B_2_O_3_-V_2_O_5_ glass system were investigated in terms of the relationship between substitution-type (i.e., V_2_O_5_) changes in typical shielding behaviors of glass system. Previously, Kilic [35] investigated the synthesis process and the optical, thermal, and structural properties of four different glasses based on zinc borate containing V_2_O_5_. His results showed that V_2_O_5_ enabled the glass-transition temperature of a sample to fall from 553 °C to 531 °C. Moreover, density declined with a rise in the quantity of V_2_O_5_. Volume improved as the V_2_O_5_ concentration increased. Compared with the earlier calculation, the refractive index improved. Their large refractive indices suggest that they may be strong optical materials that could be used in optical structures needing a high refractive index. These glasses may be used in many areas of optics, but in some sense are often innovative materials for solar-energy systems due to their semiconducting properties. However, those promising findings are worthy of continuous investigations of aspects such as resistance competencies against gamma rays. In fulfilment of international safety requirements, the value of alternate shielding materials is growing. Therefore, this paper simply focused on examining the interaction between nuclear-radiation-shielding properties in V_2_O_5_-reinforced ternary zinc borate glasses. This is because of glass materials’ potential as attenuator in industrial, medical, and research ionizing-radiation facilities. It is well known that material density has a remarkable impact on gamma-ray-shielding properties [36,37]. Therefore, the importance of traditional shields such as concrete and lead (Pb) is obvious in ionizing-radiation facilities. Since the glasses have been widely reported as alternative shields instead of the aforementioned conventional shields, this study aimed to characterize the material properties of a ZnO-B_2_O_3_-V_2_O_5_ system along with its gamma-shielding properties.

M.A. Tunes et al. developed a computational model of a compact pressurized-water nuclear reactor to investigate the use of innovative materials to enhance the biological-shielding effectiveness. They used MCNP and GEM/EVENT codes to simulate the behavior of several materials and the shielding thickness for gamma and neutron radiation [38]. J. Kaewkhao et al. investigated the mass attenuation coefficients and shielding parameters of borate-based glasses involving Bi_2_O_3_ and BaO. They established that the increment of the mass-attenuation coefficients was a function of the Bi_2_O_3_, BaO, and PbO contents. Furthermore, the half-value layers of investigated glasses were more favorable than normal concretes and marketable window glasses. These results reflect that the Bi-based glass can replace Pb in radiation-shielding glass [39]. P. Limkitjaroenporn et al. prepared lead sodium borate glasses via melt-quenching and explored their optical, physical, structural, and gamma-ray-shielding properties. They recorded that the gamma-ray-shielding properties increased with an increase in PbO concentration [40]. K. Kirdsiri et al. measured the radiation-shielding and optical properties of bismuth lead silicate and barium silicate glasses. They observed that total mass-attenuation coefficients of the glasses at 662 keV were enhanced by the increment of Bi2O3 and PbO, which elevated the photoelectric absorption in the glass networks [41]. S. Kaewjang et al. fabricated and investigated the (80−x)B_2_O_3_-10SiO_2_-10CaO-xGd_2_O_3_ (where x = 15, 20, 25, 30, and 35 mol%) for their radiation-shielding, physical, and optical properties. They found that the experimental values of the mass-attenuation coefficients, effective atomic number, and effective electron densities of the glasses increased with the increasing of Gd2O3 concentration, and also with the increasing of photon energy from 223 to 662 keV [42]. N. Chanthima et al. investigated the impacts of BaO on the physical properties of zinc borate-based glasses. They recorded that the radiation-shielding parameters were enhanced as a function of BaO, and the decrement of γ-ray energy [43]. K. Boonin et al. synthesized and inspected zinc barium tellurite glasses for their radiation-defense mechanism and structural behavior. They found that the effective atomic number and effective electron density decreased with the increase in γ-ray energies, which is in a good agreement with theoretical values attained using Geant4 and WinXCOM [44]. W. Cheewasukhanont et al. studied the radiation-shielding parameters of bismuth borosilicate-based glasses. The outcomes revealed that the density of the glasses increased with the increase of Bi_2_O_3_ content, while the particle size did not account for the density. They found that the radiation-shielding parameters of these glasses were enhanced over those for traditional glass windows and for some types of concrete [45]. S. Kaewjaeng et al. prepared and investigated (80−x)B_2_O_3_:10SiO_2_:10CaO:xLa_2_O_3_ glass (where x = 10, 15, 20, 25, and 30 mol%) for x-ray-shielding, physical, and optical properties. The found that the half-value layer and tenth-value layer of the glass samples tended to decrease when the kVp of an x-ray instrument decreased and La_2_O_3_ concentrations increased [46]. F. H. ElBatal et al. studied UV-Vis and FTIR absorption spectra of some prepared undoped and NdF3-doped borophosphate glasses, with varying dopant contents before and after gamma irradiation [47]. H. ElBatal et al. prepared undoped and CuO-doped lithium phosphate, lead phosphate, and zinc phosphate glasses. The measured the UV–VIS and infrared absorption spectra of the prepared samples before and after successive gamma irradiation [48].

This study aims to discuss the potential effects of V_2_O_5_ substitution on the nuclear-radiation-shielding properties of ZnO-B_2_O_3_-V_2_O_5_ glasses. The investigated radiation-attenuation parameters can be ordered as: linear-attenuation coefficients (LACs), mass-attenuation coefficients (MACs), effective electron density (N_eff_), half-value layer (T_1/2_), exposure buildup factor (EBF), energy-absorption buildup factor (EABF), tenth-value layer (TVL), mean free path (λ), and effective atomic number (Z_eff_). The obtained outcomes from the current investigation could be useful to understand the direct impact of glass structure, glass density, and replacement type on the radiation-shielding properties of ZnO-B_2_O_3_-V_2_O_5_ glasses. In this research report, all the investigated parameters will be discussed in terms of synergistic effect of substitution and changes in the resistance behavior against gamma radiation.

## 2. Materials and Methods

### 2.1. Behavioral Changes in ZnO-B_2_O_3_-V_2_O_5_ Glasses

Figure 1 shows that the fabricated glasses samples encoded as ZBV1, ZBV2, ZBV3, and ZBV4 based on a (100−x)(0.6ZnO–0.4B_2_O_3_)(x)(V_2_O_5_) (x = 1, 2, 3, 4 mol%) system. Glass densities and elemental compositions are presented in Table 1. As can be seen in Table 1, the values for the glass samples gradually declined from 3.392 g/cm^3^ to 3.329 g/cm^3^ with rising concentrations of V_2_O_5_ [35]. On the other hand, the molar volume values rose linearly from 22.910 cm^3^/mol to 24.289 cm^3^/mol. Therefore, a smooth decrement in gamma-ray-attenuation competencies from ZBV1 to ZBV4 can be expected to be a potential consequence.

### 2.2. MCNPX Monte Carlo Simulations

MCNPX general-purpose Monte Carlo code was implemented for gamma-ray-transmission simulations in the current investigation. Accordingly, mass-attenuation coefficients (MACs) of the (100−x)(0.6ZnO–0.4B_2_O_3_)(x)(V_2_O_5_) (x = 1, 2, 3, 4 mol%) glass system was determined. As a first step in the INPUT file, cell cards and surface cards were prepared using elemental mass fractions (wt %) and material densities (Table 1) of ZBV1, ZBV2, ZBV3, and ZBV4 glasses in the 0.015–10 MeV photon-energy range. Next, MAC values obtained from WinXcom [49] and MCNPX [50,51,52] were compared. The overall MAC values were largely stable regarding variations in incident photon energy. Despite this, smooth numerical differences were reported between WinXcom and the Monte Carlo code (Table 2).

This can be clarified by the form of processes and physics used in the Monte Carlo simulation and random event generator for a radiation-transfer method, whereas WinXcom is a mechanism that utilizes mathematical improvement for direct determination of MACs. It is worth noting that the MCNPX does not explicitly record MAC values. In some instances, some data processing is achieved by looking at performance data. Simulations were created by integrating various details, including information such as input files, cell cards, and source. The glass samples were measured in terms of their elemental mass fractions (weight per unit length), and dimensions (cm). The gamma-ray-emitting systems’ total geometry can be seen in Figure 2. In this step, the F4 Tally mesh was directly utilized for detection of attenuated gamma rays. In this tally mesh, the total photon flux in a cell was determined. Finally, it is worth mentioning that a point radioactive isotope was utilized as a source for incident gamma rays. The simulation process was repeated from 0.015 to 15 MeV for each glass sample. Moreover, a well-known variance-reduction technique known as tracking optimization was utilized. To increase simulation efficiency, neutron and electron tracking were set as off, and only photon tracking was allowed in the cell definition (i.e., IMP: P).

## 3. Results and Discussions

To determine the nuclear-radiation-shielding dynamics of V_2_O_5_-reinforced ZnO-B2O3 glasses, in silicon Monte Carlo simulation studies were performed. Accordingly, four separate glasses encoded as ZBV1, ZBV2, ZBV3, and ZBV4 based on a (100−x)(0.6ZnO-0.4B_2_O_3_)(x)(V_2_O_5_) (x = 1, 2, 3, 4 mol. %) glass system was checked for their potentials as possible materials for nuclear-radiation-shielding implementations. In this study, we used Monte Carlo simulations and Phy-X PSD [53] code for determination of shielding parameters. First, linear-attenuation coefficients (LACs) were determined between 0.015–15 MeV gamma photon energy. Figure 3 depicts the variation in the linear attenuation coefficients (µ) against photon energy for all glasses. Figure 3 shows that the influence of the photoelectric effect, Compton scattering, and pair production processes on the LAC was variable depending on the incident energy zones (i.e., low, middle, and high energy). This finding follows as a result of the association of radiation with matter. At low energy values, the LAC is lost due to photo adsorption. Compton scattering accounted for the majority of the energy difference at the mid-energy area, while the lowest LAC was reported in the ZBV4 sample. In other words, the ZBV4 sample with the highest amount of V_2_O_5_ additive showed the lowest LAC values. In contrast, the ZBV1 sample showed the highest LAC values. In the low-energy region, LAC values were quite similar, whereas in the high-energy region, the differences were slightly higher. The importance of density and direct LAC relationship with density were dominant factors (Figure 3). The mass-attenuation coefficient (MAC) of a material is a specific parameter that can be used in terms of density-independent categorizations of studied shielding materials. Figure 4 illustrates the variations in the mass-attenuation coefficient (µ_m_) against incident photon energy for all glasses at 0.015–15 MeV. The chemical structure of the attenuator glass altered the rate of variation of the calculated MAC values. The mathematical analysis of the MAC data show patterns in distinct regions. In the low-energy condition, in which the photoelectric effect is important, the absorption dramatically decreased. Compton-scattering superiority showing the total decrease from the MAC at the mid-energy region. The ZBV1 sample displayed very high atomic concentrations at all incident photon energies. The situation in both the low- and high-energy areas can be clarified by the presence of the highest volume of zinc (*Z* = 30) in the glass structure, since it has a significant atomic number among the utilized composition elements (i.e., B, O, V, Zn).

HVL, which stands for “half-value layer” (T_1/2_), is a valuable amount when measuring the thickness of a shielding material in order to minimize gamma rays. So, a smaller HVL may serve as a measure of a material’s capacity to shield incident gamma rays from a primary source. Using the calculated LAC values, the HVLs of the studied glasses were calculated. Figure 5 demonstrates the association between photon energy and T_1/2_ for all glass samples. In the low-energy spectrum, calculations of HVL values were reported smaller. This is a predictable result for shielding in materials that low-intensity gamma rays cannot move through almost completely. To learn more about the efficiency of the studied materials as gamma-ray absorbers, we found it useful to compare them with some of the materials most commonly used as shields against gamma rays. The HVL value for the best glass sample (ZBV4 = 8.13915 cm) at 15 MeV was compared with other shielding materials (lead= 1.08069 cm, ordinary concrete = 13.89992 cm, HSC = 11.63975 cm, ILC = 9.40269 cm, IL = 7.81848 cm, G1 = 15.25284 cm, and G2 = 10.53327 cm) glasses [54,55]. The comparison shows that the ZBV4 glass came third after lead, and with IL outperforming the other shield types. Considering its other distinguishing features, such as lack of toxicity risk and lower weight, this glass is recommended as the ideal choice in many applications that require gamma-ray shields with specific properties.

However, Figure 5 shows that the HVL values were greater in the mid- as well as the high-energy field. This was attributed to the penetration property of gamma rays and how it changes with higher energy levels. Considering the direct impact of material densities, which were reported as 3.392 g/cm^3^, 3.371 g/cm^3^, 3.339 g/cm^3^, and 3.329g/cm^3^ for ZBV1, ZBV2, ZBV3, and ZBV4, respectively, the HVL values were changed in similar order. For example, HVL values of 1.109 cm HVL_ZBV1_ < 1.121 cm HVL_ZBV2_ < 1.136 HVL_ZBV3_ < 1.143 cm HVL_ZBV4_ were reported at 0.15 MeV. The initial difference in HVL values was 0.034 cm at 0.015 MeV. To confirm the previously mentioned interpretation of the changing trend of HVL values depending on energy region, it is favorable to order those HVL values in the high-energy region. Thus, 7.948 cm HVL_ZBV1_ < 8.011 cm HVL_ZBV2_ < 8.101 cm HVL_ZBV3_ < 8.139 cm HVL_ZBV4_ were also reported at 15 MeV. In this case, the difference in HVL values was 0.191 cm at 0.015 MeV. Therefore, one can say that the negative effect of V_2_O_5_ on HVL values was slightly lower at low energies. Accordingly, the aforementioned reinforcement type negativity can be tolerated at low energy values, considering the positive effects on optical, structural, and thermal properties. The effect of the mean free path (λ) is critical in the capacity of shielding materials to defend against gamma radiation. Glass samples were analyzed in terms of λ values, and details are summarized in Figure 6. The λ values can fluctuate, similar to the shifting patterns in the HVL. The λ values of ZBV1 have been recoded as the minimum in the investigated energy range. Another main element to remember is the tenth-value layer (TVL). The TVL element is another critical criterion to verify the thickness of shields, minimizing gamma rays’ strength to 1/10 of their original intensity. The TVLs of the studied samples were calculated, and a graph was drawn showing their importance against incident photon energy (Figure 7). In the lower-energy ranges, the TVL values were lower. Similar to the changing trend in HVL values, TVL values were registered as 3.685 cm HVL_ZBV1_ < 3.722 cm HVL_ZBV2_ < 3.772 cm HVL_ZBV3_ < 3.797 cm HVL_ZBV4_ at 0.15 MeV. The effective atomic number (Z_eff_) of a substance used for gamma-ray exposure regulation would contribute to its photon-attenuation properties. The variation of effective atomic number (Z_eff_) against photon energy for all glasses is shown in Figure 8.

The Z_eff_ values for ZBV1 glass were the best. For example, a Z_eff_ value of 14.01 was found for the ZBV1 sample at 0.15 MeV. However, the maximum Z_eff_ values recorded for samples were 28.42, 28.30, 28.19, and 28.07 for ZBV1, ZBV2, ZBV3, and ZBV4, respectively. The average photon flux was measured in the detection field by using an F4 tally mesh (Figure 2). Then, an absorption coefficient was used to compensate for a scattering of the radiation. Excess radiation required the cumulative measurement of radiation buildup. The attenuation component was multiplied by the photon’s attenuation rate to measure the photon’s overall attenuation. The moderator attenuates the penetration of photons of varying intensities such that they both achieve the same reception. In this case, buildup factors can be classified as exposure buildup factor (EBF) and energy-absorption buildup factor (EABF).

In this analysis, a G-P fitting method was utilized for determination of EBF and EABF values. The fitting parameters’ calculated numerical values are displayed in Table 3, Table 4, Table 5 and Table 6 along with equivalent atomic numbers (Zeq). Figure 9a–d and Figure 10a–d illustrate the measured EBF and EABF values for studied glasses at various mean free paths from 0.6 to 40 mfp. Figure 9 and Figure 10 express the concentration of the different layers of EBF and EABF owing to irradiation by gamma rays. We shall explain the changing trend of EBF and EABF values at different energy values. Due to the photoelectric influence being neglected in regions with large atomic numbers, one may see smooth increments in the first region (Table 3, Table 4, Table 5 and Table 6). In addition, the third section of the sample was the most interesting during pair growth due to its absorption processes, which resulted in the decline in its value. Figure 9 and Figure 10 depict the changes in EBF and EABF values of ZBV1, ZBV2, ZBV3, and ZBV4 glass samples for different mean free path (mfp) values (i.e., 0.5, 1, 2, 3, 4, 5, 6, 7, 8, 9, 10, 15, 20, 25, 30, 35, and 40 mfp). These figures demonstrate the reliance of the EBF and EABF values upon the glass structure. The obtained EBF and EABF values showed that ZBV1 had the lowest EBF and EABF levels, and was superior to all other glasses.

Figure 11 and Figure 12 show the differing energy-absorption buildup factors (EABF) and exposure buildup factors (EBF) at 0.4 MeV along with 10, 20, 30, and 40 mfp values for all glasses. It can be seen that the ZBV1 sample showed the minimum EBF and EABF values at the studied mfp values. From ZBV1 to ZBV4, EBF and EABF values showed an increasing trend. This can be explained by decreasing gamma-ray-attenuation competencies with a V_2_O_5_ reinforcement amount from 1 to 4 mol.%. As a last analysis of EBF and EABF values, Figure 13 shows the variations in the energy-absorption buildup factor (EABF) and exposure buildup factor (EBF) against effective atomic number (Z_eff_) for all glasses at 1 MeV/5 mfp. This is another sign of the synergistic effect of V_2_O_5_ reinforcement, which caused a decrement in densities of glasses, and in gamma-ray-shielding competencies accordingly. As illustrated in Figure 13, EBF values increased gradually from ZBV1 to ZBV4. This trend was a natural consequence of variations in effective atomic numbers, which was discussed in previous sections.

## 4. Conclusions

Glasses doped with transition-metal oxides were reported to show semiconducting properties in many experimental data due to their multivalid states. Furthermore. boron oxide-based glasses are preferred since they can be synthesized easily, and due to their many characteristic optical and thermal properties. In fact, since zinc borate oxide glasses have a broad spectrum of formation range, they have been the hosts for dopant many times. Zinc borate glasses with a high ZnO ratio were doped with vanadium oxide at varying ratios and constituted the study samples. Increasing the amount of V_2_O_5_ within the structure resulted in a negative effect, contrary to what was expected in a previous study, and this was valid for the radiation properties observed in this study. V_2_O_5_ replaced B_2_O_3_ and ZnO. Though vanadium is an element that is heavier than boron. it is lighter than zinc. The domination of ZnO concentration in the composition compared to B_2_O_3_ made the ZnO substitution with V_2_O_5_ more dominant, leading to a decrease in density. Since density has a significant role in data obtained from radiation calculations, the negative effect was observed. The first outcome can be associated with the relationship of density variation and gamma-ray-attenuation properties, since the density values of the samples were changed with the V_2_O_5_ additive. Considering the obtained results for gamma-ray-attenuation properties, it can be concluded that V_2_O_5_ reinforcement had an adverse impact against gamma rays. This situation was obtained for all the gamma-ray-attenuation properties such as LAC, MAC, HVL, TVL, mfp, Z_eff_, EBF, and EABF. Some of the numerical values were discussed in previous sections. Among the investigated glasses, ZBV1 was reported to have the highest gamma-ray-attenuation properties. Therefore, it can be said that ZBV1 would attenuate incident gamma rays at the maximum level. It can be concluded that some other investigations, such as on mechanical properties acquired upon utilization, should be performed.

## Figures and Tables

**Figure 1 materials-14-01158-f001:**
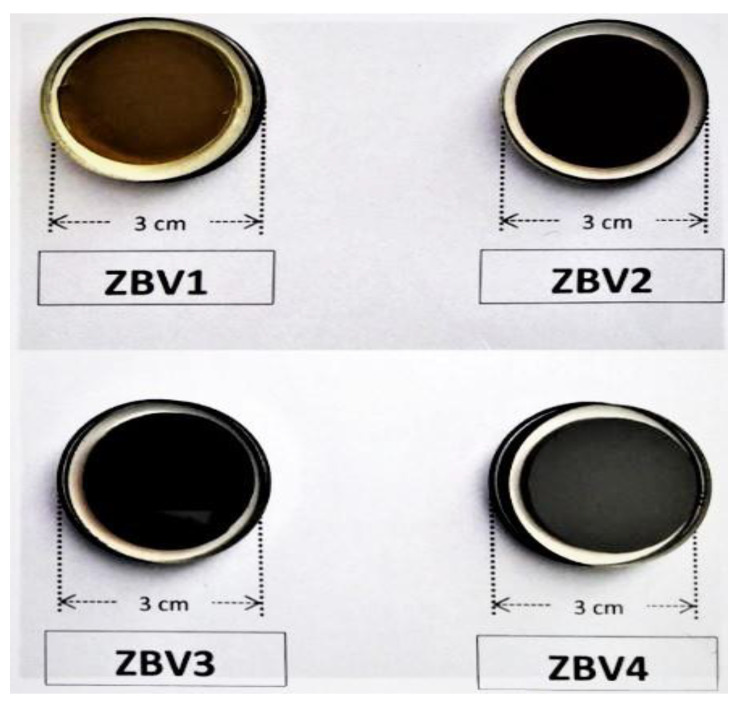
Studied glass samples [35] and their physical/optical appearances.

**Figure 2 materials-14-01158-f002:**
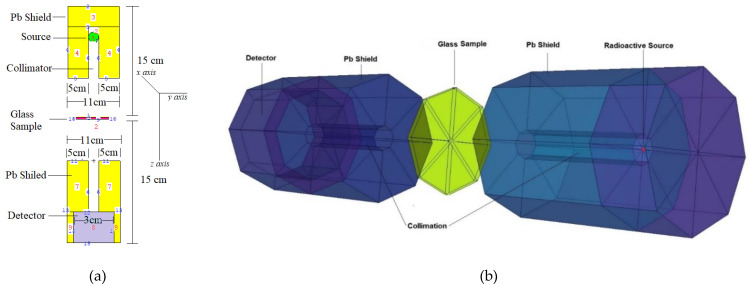
MCNPX simulation setup for gamma-ray-transmission studies: (**a**) 2D view with dimensions, (**b**) 3D view of setup obtained from MCNPX Visual Editor.

**Figure 3 materials-14-01158-f003:**
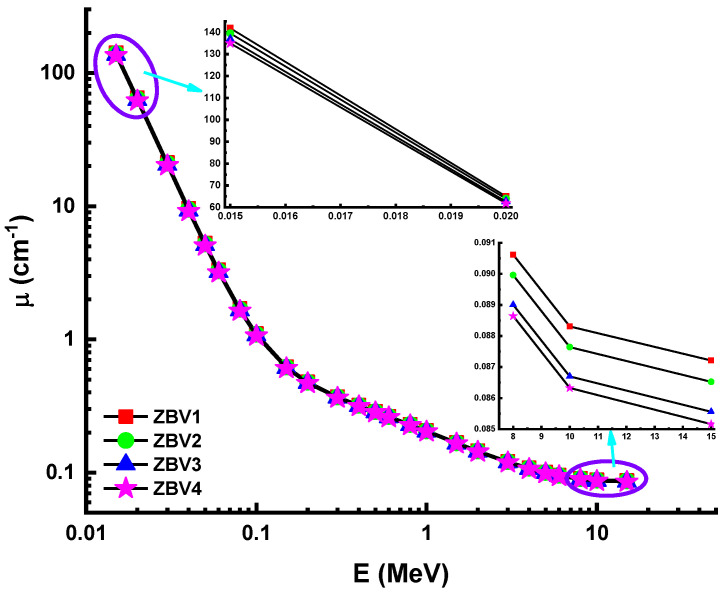
Variation of linear-attenuation coefficient (µ) against photon energy for all glasses.

**Figure 4 materials-14-01158-f004:**
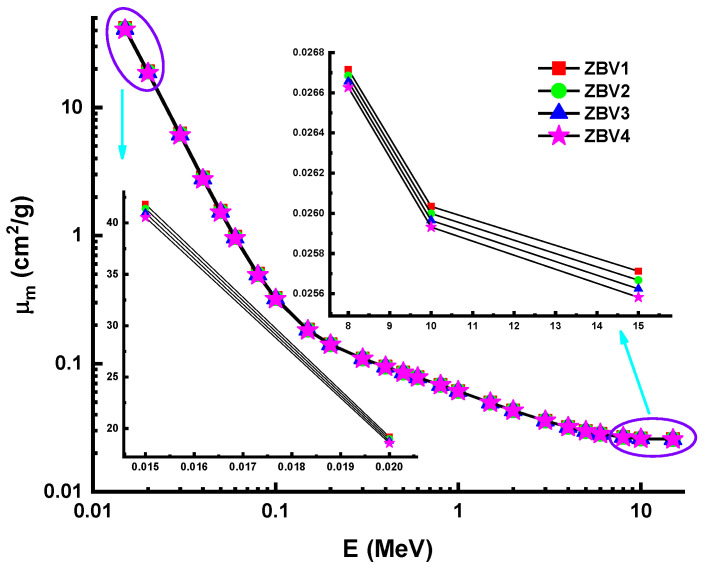
Variation of mass-attenuation coefficient (µ_m_) against photon energy for all glasses.

**Figure 5 materials-14-01158-f005:**
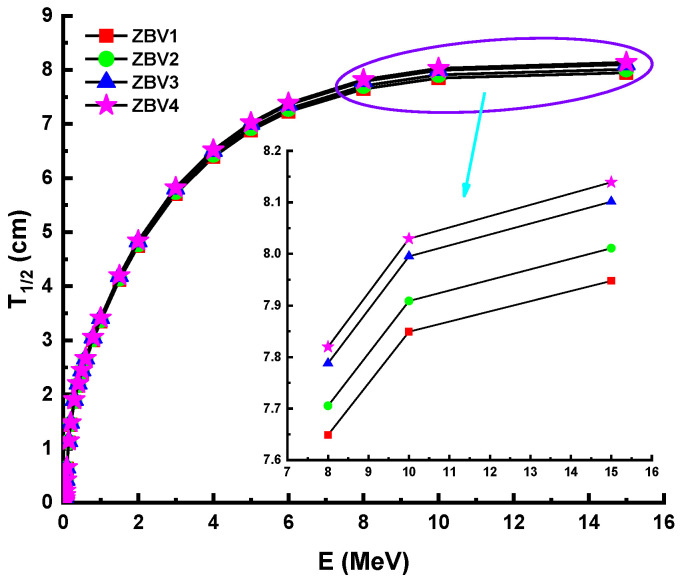
Variation of half-value layer (T_1/2_) against photon energy for all glasses.

**Figure 6 materials-14-01158-f006:**
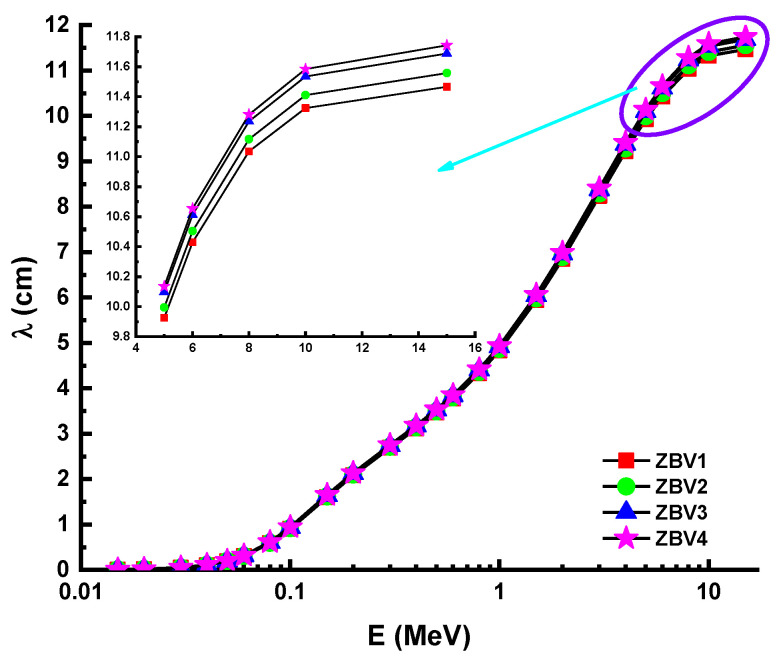
Variation of mean free path (λ) against photon energy for all glasses.

**Figure 7 materials-14-01158-f007:**
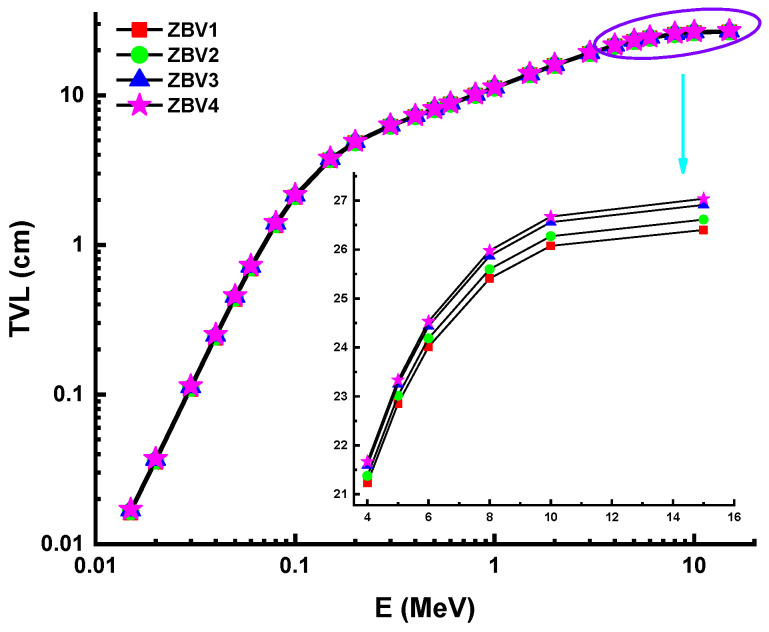
Variation of tenth-value layer (TVL) against photon energy for all glasses.

**Figure 8 materials-14-01158-f008:**
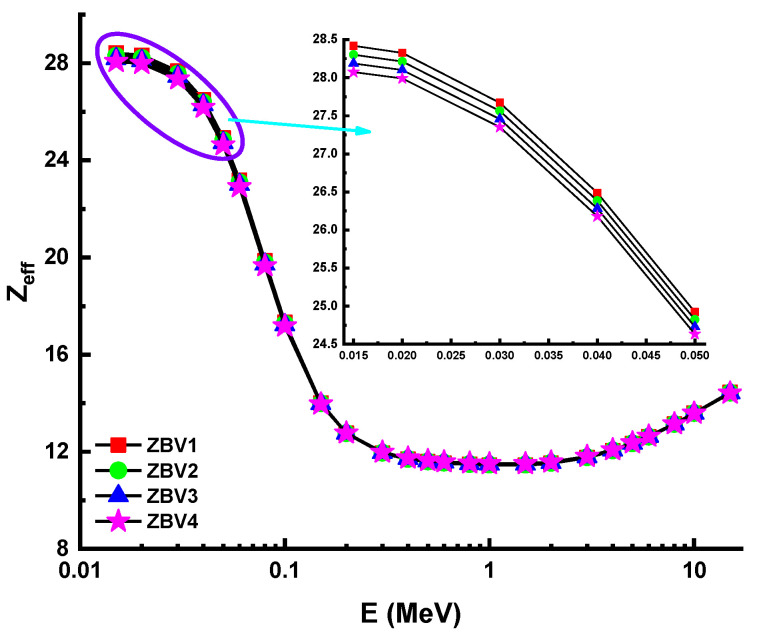
Variation of effective atomic number (Z_eff_) against photon energy for all glasses.

**Figure 9 materials-14-01158-f009:**
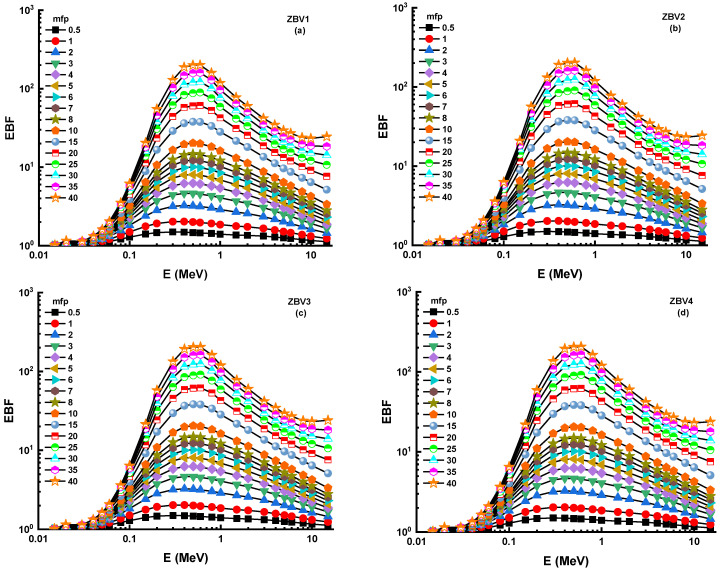
(**a**–**d**): Variation of exposure buildup factor (EBF) against photon energy for all glasses.

**Figure 10 materials-14-01158-f010:**
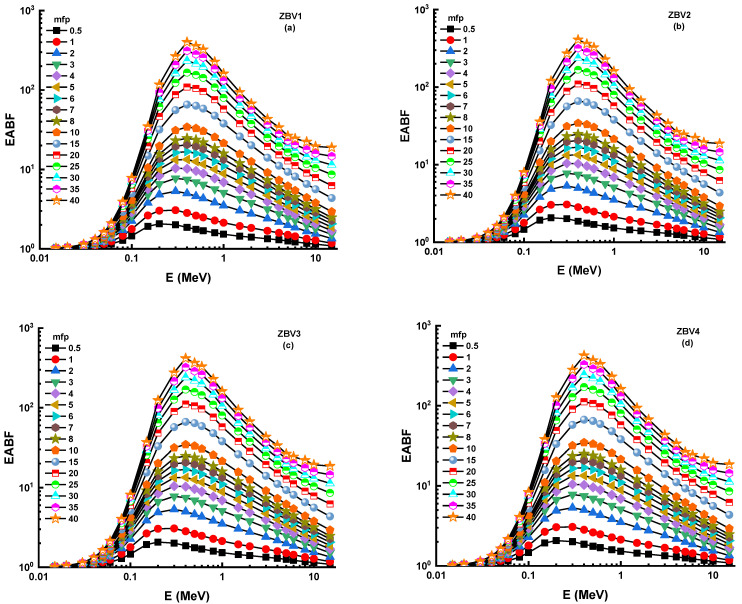
(**a**–**d**): Variation of energy-absorption buildup factor (EABF) against photon energy for all glasses.

**Figure 11 materials-14-01158-f011:**
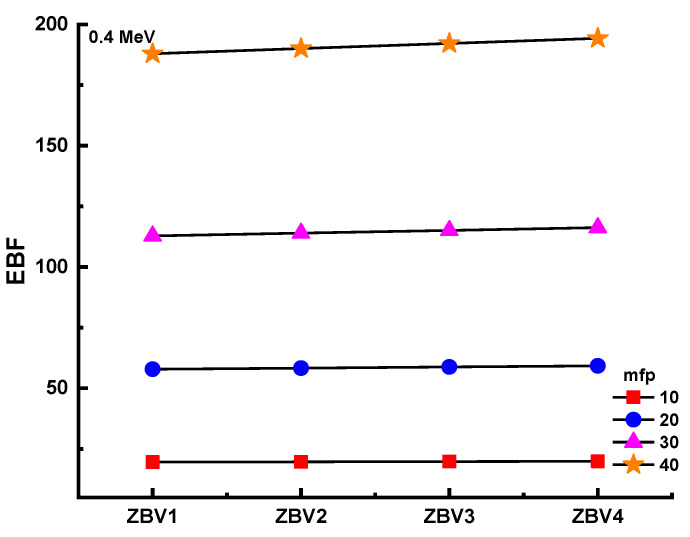
Variation of exposure buildup factor (EBF) against glass compositions.

**Figure 12 materials-14-01158-f012:**
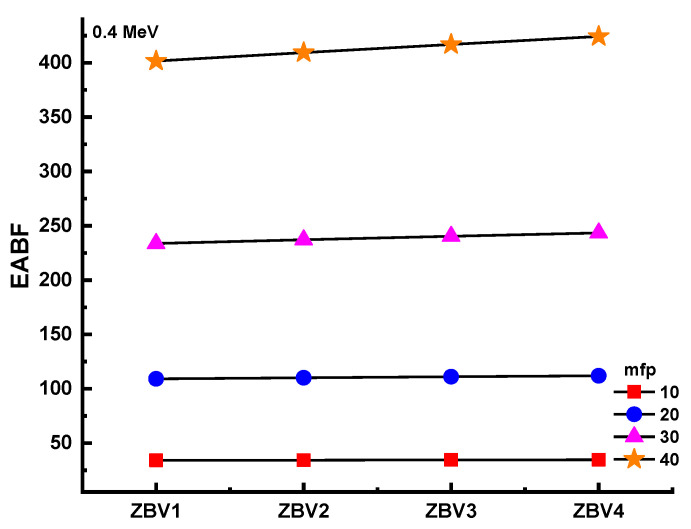
Variation of energy-absorption buildup factor (EABF) against glass compositions.

**Figure 13 materials-14-01158-f013:**
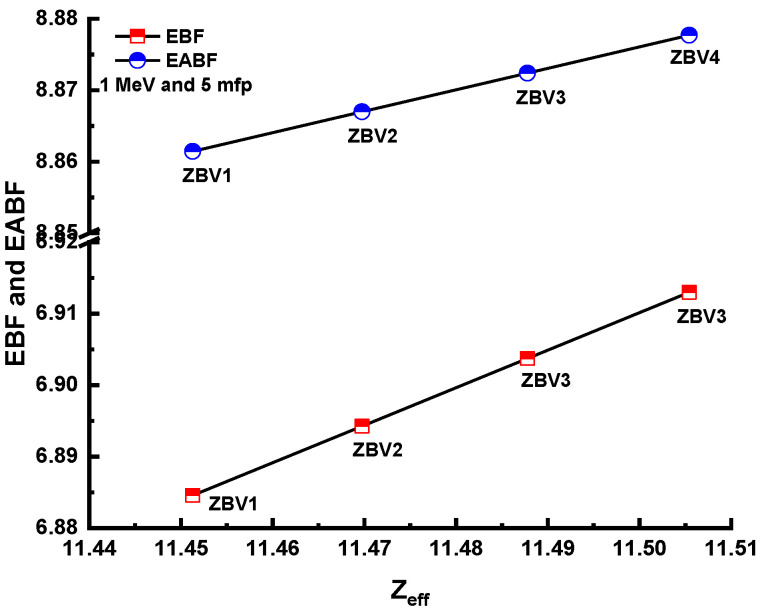
Variation of energy-absorption buildup factor (EABF) and exposure buildup factor (EBF) against effective atomic number (Z_eff_) for all glasses.

**Table 1 materials-14-01158-t001:** Chemical compositions and density for glass samples.

Glass Code	mol%			wt%				Density (g/cm^3^)
	ZnO	B_2_O_3_	V_2_O_5_	B	O	V	Zn	
ZBV1	59.4	39.6	1	0.110149663	0.377094852	0.013108056	0.499647429	3.392
ZBV2	58.8	39.2	2	0.1075809	0.378557769	0.025866008	0.487995323	3.371
ZBV3	58.2	38.8	3	0.105079843	0.379982128	0.038287697	0.476650332	3.339
ZBV4	57.6	38.4	4	0.102643849	0.381369433	0.050386245	0.465600473	3.329

**Table 2 materials-14-01158-t002:** Mass-attenuation coefficients (cm^2^/g) of the studied glass samples obtained using the MCNPX code and Phy-X PSD program.

Energy (MeV)	ZBV1		ZBV2		ZBV3		ZBV4	
	*Phy-X PSD*	*MCNPX*	*Phy-X PSD*	*MCNPX*	*Phy-X PSD*	*MCNPX*	*Phy-X PSD*	*MCNPX*
0.015	41.8266	43.2654	41.3902	42.8624	40.9654	41.2416	40.5516	41.0625
0.02	19.1754	19.6521	18.9681	19.5124	18.7662	18.8126	18.5695	18.7250
0.03	6.2674	6.3124	6.1978	6.3004	6.1300	6.1526	6.0640	6.1236
0.04	2.8399	2.8524	2.8087	2.8324	2.7782	2.7816	2.7485	2.7628
0.05	1.5617	1.5721	1.5451	1.5629	1.5289	1.523	1.5131	1.5321
0.06	0.9800	0.9936	0.9701	0.9824	0.9605	0.972	0.9512	0.9626
0.08	0.5032	0.5092	0.4989	0.5054	0.4947	0.5023	0.4906	0.4926
0.1	0.3261	0.3295	0.3238	0.3286	0.3217	0.3251	0.3195	0.3198
0.15	0.1842	0.1901	0.1835	0.1882	0.1828	0.1862	0.1821	0.1832
0.2	0.1417	0.1421	0.1414	0.141	0.1411	0.1406	0.1408	0.1409
0.3	0.1096	0.111	0.1095	0.1101	0.1094	0.1091	0.1093	0.1089
0.4	0.0946	0.0952	0.0946	0.095	0.0945	0.0947	0.0945	0.0946
0.5	0.0851	0.086	0.085	0.0856	0.085	0.0854	0.085	0.0852
0.6	0.0781	0.0792	0.078	0.079	0.078	0.0782	0.078	0.0781
0.8	0.068	0.0695	0.068	0.0692	0.068	0.0688	0.068	0.0684
1	0.061	0.0624	0.0609	0.0621	0.0609	0.0619	0.0609	0.0612
1.5	0.0496	0.0501	0.0496	0.0499	0.0496	0.0495	0.0496	0.0494
2	0.043	0.0446	0.043	0.0439	0.043	0.0436	0.043	0.0434
3	0.0358	0.0363	0.0358	0.036	0.0358	0.0359	0.0358	0.0356
4	0.032	0.0334	0.032	0.0333	0.0319	0.0331	0.0319	0.0325
5	0.0297	0.0309	0.0297	0.0305	0.0297	0.0302	0.0296	0.0301
6	0.0283	0.0286	0.0282	0.0284	0.0282	0.0283	0.0282	0.0282
8	0.0267	0.0271	0.0267	0.027	0.0267	0.0268	0.0266	0.0267
10	0.026	0.0263	0.026	0.0261	0.026	0.026	0.0259	0.0259
15	0.0257	0.0262	0.0257	0.026	0.0256	0.0259	0.0256	0.0257

**Table 3 materials-14-01158-t003:** (EBF and EABF) G–P fitting coefficients for the ZBV1 glass sample.

Energy (MeV)	Z_eq_	G–P Fitting Parameters for EBF					G–P Fitting Parameters for EABF				
		a	b	c	d	X_k_	a	b	c	d	X_k_
0.015	22.43	0.230	1.006	1.018	0.227	0.223	1.006	1.011	0.217	7.920	−0.223
0.020	22.78	0.398	1.014	0.287	−0.348	0.283	1.014	0.314	−0.236	14.711	0.283
0.030	23.21	0.206	1.041	0.373	−0.231	0.246	1.040	0.337	−0.172	16.253	0.246
0.040	23.47	0.243	1.086	0.347	−0.125	0.238	1.085	0.350	−0.127	13.895	0.238
0.050	23.65	0.225	1.145	0.380	−0.129	0.237	1.154	0.360	−0.134	14.335	0.237
0.060	23.80	0.203	1.211	0.422	−0.113	0.216	1.240	0.398	−0.126	14.724	0.216
0.080	24.00	0.168	1.358	0.502	−0.092	0.174	1.463	0.482	−0.098	15.336	0.174
0.100	24.13	0.129	1.495	0.600	−0.073	0.156	1.772	0.541	−0.094	15.276	0.156
0.150	24.32	0.065	1.763	0.796	−0.043	0.125	2.632	0.657	−0.101	14.589	0.125
0.200	24.43	0.023	1.919	0.960	−0.030	0.061	3.026	0.856	−0.062	13.065	0.061
0.300	24.55	−0.016	2.024	1.131	−0.017	−0.001	3.065	1.086	−0.030	11.983	−0.001
0.400	24.61	−0.031	2.030	1.210	−0.013	−0.029	2.828	1.204	−0.013	14.934	−0.029
0.500	24.65	−0.041	2.000	1.253	−0.008	−0.041	2.624	1.256	−0.008	11.683	−0.041
0.600	24.67	−0.044	1.969	1.264	−0.005	−0.043	2.484	1.266	−0.006	12.436	−0.043
0.800	24.70	−0.042	1.917	1.250	−0.006	−0.044	2.273	1.262	−0.007	9.857	−0.044
1.000	24.70	−0.050	1.853	1.258	0.014	−0.051	2.130	1.265	0.013	17.995	−0.051
1.500	22.35	−0.042	1.777	1.205	0.012	−0.043	1.937	1.210	0.013	15.494	−0.043
2.000	19.86	−0.030	1.738	1.147	0.008	−0.029	1.834	1.144	0.007	17.096	−0.029
3.000	18.94	−0.004	1.658	1.047	−0.011	−0.008	1.685	1.057	−0.006	12.383	−0.008
4.000	18.68	0.008	1.579	1.003	−0.017	0.014	1.590	0.984	−0.023	12.087	0.014
5.000	18.56	0.014	1.510	0.982	−0.021	0.021	1.501	0.962	−0.033	14.147	0.021
6.000	18.48	0.021	1.462	0.962	−0.026	0.027	1.435	0.944	−0.034	12.913	0.027
8.000	18.40	0.032	1.381	0.935	−0.036	0.034	1.339	0.928	−0.034	12.212	0.034
10.000	18.35	0.039	1.319	0.922	−0.042	0.044	1.281	0.903	−0.047	13.903	0.044
15.000	18.31	0.064	1.236	0.866	−0.065	0.035	1.176	0.944	−0.039	14.486	0.035

**Table 4 materials-14-01158-t004:** (EBF and EABF) G–P fitting coefficients for the ZBV2 glass sample.

Energy (MeV)	Z_eq_	G–P Fitting Parameters for EBF					G–P Fitting Parameters for EABF				
		a	b	c	d	X_k_	a	b	c	d	X_k_
0.015	22.35	0.222	1.006	1.004	0.223	6.425	0.215	1.006	0.998	0.213	7.978
0.020	22.70	0.392	1.014	0.291	−0.341	11.131	0.282	1.014	0.316	−0.234	14.615
0.030	23.13	0.206	1.041	0.373	−0.228	21.387	0.245	1.040	0.338	−0.171	16.167
0.040	23.39	0.242	1.087	0.347	−0.126	12.627	0.238	1.086	0.351	−0.127	13.894
0.050	23.57	0.225	1.147	0.381	−0.129	14.059	0.236	1.156	0.361	−0.134	14.333
0.060	23.71	0.202	1.214	0.423	−0.113	14.198	0.215	1.243	0.399	−0.126	14.725
0.080	23.91	0.168	1.362	0.503	−0.092	14.438	0.173	1.468	0.484	−0.097	15.347
0.100	24.04	0.129	1.500	0.602	−0.073	14.167	0.154	1.780	0.544	−0.094	15.290
0.150	24.23	0.064	1.768	0.798	−0.042	14.004	0.124	2.644	0.661	−0.100	14.567
0.200	24.34	0.022	1.924	0.963	−0.030	13.066	0.060	3.034	0.860	−0.062	13.062
0.300	24.46	−0.016	2.027	1.133	−0.017	11.487	−0.002	3.067	1.090	−0.030	11.952
0.400	24.52	−0.031	2.032	1.212	−0.013	10.506	−0.029	2.829	1.207	−0.012	15.154
0.500	24.56	−0.041	2.002	1.254	−0.008	8.420	−0.041	2.624	1.259	−0.008	11.778
0.600	24.58	−0.044	1.971	1.265	−0.004	11.502	−0.044	2.484	1.267	−0.005	12.591
0.800	24.61	−0.043	1.918	1.252	−0.005	10.254	−0.044	2.273	1.263	−0.007	9.970
1.000	24.61	−0.050	1.854	1.258	0.014	18.992	−0.051	2.130	1.266	0.013	17.965
1.500	22.26	−0.042	1.777	1.205	0.012	15.823	−0.043	1.937	1.210	0.013	15.498
2.000	19.78	−0.030	1.738	1.147	0.008	17.345	−0.029	1.834	1.145	0.007	17.043
3.000	18.88	−0.004	1.658	1.048	−0.011	11.312	−0.008	1.686	1.056	−0.006	12.417
4.000	18.62	0.008	1.579	1.003	−0.017	11.339	0.014	1.590	0.984	−0.022	12.128
5.000	18.50	0.014	1.510	0.982	−0.021	13.395	0.021	1.501	0.963	−0.032	14.300
6.000	18.42	0.021	1.462	0.963	−0.026	13.579	0.027	1.435	0.944	−0.034	12.928
8.000	18.34	0.031	1.381	0.936	−0.035	13.418	0.034	1.339	0.927	−0.034	12.221
10.000	18.30	0.039	1.319	0.921	−0.042	13.453	0.044	1.281	0.905	−0.046	13.904
15.000	18.25	0.066	1.237	0.862	−0.066	13.770	0.035	1.176	0.944	−0.038	14.494

**Table 5 materials-14-01158-t005:** (EBF and EABF) G–P fitting coefficients for the ZBV3 glass sample.

Energy (MeV)	Z_eq_	G–P Fitting Parameters for EBF					G–P Fitting Parameters for EABF				
		a	b	c	d	X_k_	a	b	c	d	X_k_
0.015	22.27	0.214	1.006	0.991	0.220	6.445	0.207	1.006	0.984	0.210	8.035
0.020	22.62	0.386	1.014	0.295	−0.334	11.124	0.281	1.014	0.318	−0.232	14.520
0.030	23.04	0.207	1.041	0.373	−0.225	21.137	0.245	1.040	0.339	−0.170	16.081
0.040	23.30	0.242	1.088	0.348	−0.126	12.661	0.237	1.087	0.351	−0.127	13.892
0.050	23.48	0.225	1.148	0.381	−0.128	14.061	0.236	1.157	0.361	−0.134	14.330
0.060	23.62	0.202	1.216	0.424	−0.113	14.199	0.215	1.246	0.401	−0.125	14.726
0.080	23.82	0.167	1.366	0.505	−0.092	14.437	0.173	1.474	0.486	−0.097	15.357
0.100	23.95	0.128	1.505	0.604	−0.072	14.169	0.153	1.789	0.547	−0.093	15.303
0.150	24.14	0.063	1.774	0.801	−0.042	13.998	0.123	2.655	0.665	−0.099	14.546
0.200	24.25	0.022	1.928	0.966	−0.030	13.057	0.059	3.042	0.863	−0.061	13.059
0.300	24.37	−0.016	2.030	1.135	−0.017	11.464	−0.003	3.069	1.093	−0.030	11.921
0.400	24.43	−0.032	2.035	1.214	−0.013	10.492	−0.030	2.829	1.210	−0.012	15.370
0.500	24.47	−0.041	2.004	1.255	−0.008	8.425	−0.042	2.623	1.261	−0.007	11.871
0.600	24.49	−0.044	1.972	1.267	−0.004	11.717	−0.044	2.484	1.269	−0.005	12.743
0.800	24.52	−0.043	1.919	1.253	−0.005	10.408	−0.045	2.273	1.265	−0.006	10.080
1.000	24.52	−0.050	1.855	1.259	0.014	18.959	−0.051	2.130	1.267	0.013	17.936
1.500	22.17	−0.042	1.778	1.205	0.012	15.821	−0.043	1.937	1.211	0.013	15.502
2.000	19.71	−0.030	1.739	1.147	0.007	17.271	−0.029	1.834	1.145	0.007	16.991
3.000	18.82	−0.004	1.658	1.048	−0.010	11.333	−0.008	1.686	1.056	−0.006	12.451
4.000	18.56	0.008	1.579	1.003	−0.017	11.305	0.014	1.590	0.984	−0.022	12.169
5.000	18.44	0.014	1.511	0.982	−0.021	13.393	0.020	1.501	0.964	−0.032	14.450
6.000	18.37	0.020	1.462	0.963	−0.025	13.595	0.027	1.435	0.944	−0.034	12.943
8.000	18.29	0.031	1.381	0.936	−0.035	13.413	0.034	1.340	0.927	−0.034	12.231
10.000	18.24	0.039	1.320	0.921	−0.042	13.447	0.043	1.281	0.906	−0.046	13.905
15.000	18.20	0.067	1.239	0.858	−0.067	13.791	0.034	1.176	0.945	−0.038	14.502

**Table 6 materials-14-01158-t006:** (EBF and EABF) G–P fitting coefficients for the ZBV4 glass sample.

Energy (MeV)	Z_eq_	G–P Fitting Parameters for EBF					G–P Fitting Parameters for EABF				
		a	b	c	d	X_k_	a	b	c	d	X_k_
0.015	22.19	0.206	1.006	0.978	0.217	6.465	0.198	1.006	0.971	0.207	8.091
0.020	22.54	0.380	1.014	0.300	−0.326	11.117	0.280	1.014	0.319	−0.231	14.426
0.030	22.96	0.207	1.042	0.373	−0.223	20.895	0.245	1.041	0.339	−0.170	15.999
0.040	23.22	0.242	1.089	0.348	−0.126	12.695	0.237	1.088	0.352	−0.127	13.891
0.050	23.40	0.224	1.150	0.382	−0.128	14.062	0.236	1.159	0.362	−0.134	14.328
0.060	23.54	0.202	1.219	0.424	−0.113	14.200	0.214	1.249	0.402	−0.125	14.727
0.080	23.73	0.167	1.370	0.506	−0.091	14.435	0.172	1.479	0.488	−0.096	15.367
0.100	23.86	0.127	1.510	0.606	−0.072	14.172	0.152	1.797	0.550	−0.093	15.317
0.150	24.05	0.063	1.779	0.804	−0.042	13.993	0.121	2.666	0.669	−0.098	14.525
0.200	24.16	0.021	1.933	0.969	−0.029	13.049	0.058	3.050	0.867	−0.061	13.056
0.300	24.28	−0.017	2.033	1.137	−0.016	11.443	−0.004	3.071	1.096	−0.029	11.891
0.400	24.35	−0.032	2.037	1.215	−0.013	10.479	−0.030	2.829	1.212	−0.011	15.580
0.500	24.38	−0.041	2.006	1.256	−0.008	8.430	−0.042	2.623	1.263	−0.007	11.961
0.600	24.40	−0.045	1.974	1.268	−0.004	11.927	−0.045	2.484	1.271	−0.004	12.891
0.800	24.43	−0.043	1.920	1.254	−0.004	10.559	−0.045	2.272	1.266	−0.006	10.188
1.000	24.43	−0.050	1.856	1.259	0.014	18.927	−0.051	2.130	1.267	0.013	17.908
1.500	22.08	−0.042	1.779	1.205	0.012	15.819	−0.043	1.937	1.211	0.013	15.506
2.000	19.64	−0.030	1.739	1.147	0.007	17.198	−0.029	1.834	1.145	0.007	16.941
3.000	18.76	−0.004	1.658	1.048	−0.010	11.354	−0.008	1.687	1.056	−0.006	12.484
4.000	18.50	0.008	1.579	1.003	−0.016	11.271	0.014	1.590	0.984	−0.022	12.208
5.000	18.39	0.014	1.511	0.981	−0.021	13.392	0.020	1.501	0.965	−0.032	14.597
6.000	18.31	0.020	1.462	0.963	−0.025	13.610	0.027	1.435	0.944	−0.034	12.957
8.000	18.23	0.031	1.380	0.936	−0.035	13.409	0.035	1.340	0.926	−0.034	12.240
10.000	18.19	0.039	1.320	0.921	−0.042	13.441	0.043	1.281	0.907	−0.045	13.906
15.000	18.15	0.068	1.240	0.854	−0.068	13.812	0.034	1.176	0.946	−0.037	14.509

## Data Availability

Not applicable.

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
