# Peer review of "In-Silico Monte Carlo Simulation Trials for Investigation of V_2_O_5_ Reinforcement Effect on Ternary Zinc Borate Glasses: Nuclear Radiation Shielding Dynamics"

_materials, 2021, doi:10.3390/ma14051158_

Round 1

Reviewer 1 Report

The authors tried to investigate V2O5 effect on radiation shielding of ternary zinc borate glasses using the numerical approach. The idea is fine, but some important parts are missing or need revision in both research design and writing. Some examples are listed below, but the authors are recommended to go through the whole paper again to address any other similar issues:

1. For the simulation in 2.2, did you use any experiments to characterize glass properties as simulation input, or use theoretical values?

2. Line 141 to 156 is similar to the introduction, you may consider move this part upward and combine with the introduction. Actually, section 3 should start with simulation results directly.

3. You mentioned in line 168 to 169 that difference in LAC values are higher in the high energy region, but I do not see that clearly from figure 3. You should use same y-scales in the two zoom-in plots for a reasonable comparison.

4. Difference in figure 4 and 7 are not clear. Please add zoom-in plots like you did for other figures.

5. How did you measure the number of particles mentioned in line 236? Experimentally or theoretically? Not clear here.

6. It is suggested to put tables right after the paragraphs that first mention them, so the table and content connection will be more clear.

7. In section 2.1 you show a figure of real materials. Did you conduct any experiments for material property characterization or simulation results validation.

8. As far as I can see, the simulation procedure is quite standard and the modeling input is obtained using simple theoretical calculation. As a result, the conclusion you get (lower density leads to lower radiation shielding) is quite trivial. Can you add some more content, such as theory, a new modeling method or experiments to provide more in-depth analysis for this glass radiation shielding phenomenon?

Author Response

Reviewer 1

Comments and Suggestions for Authors

The authors tried to investigate V2O5 effect on radiation shielding of ternary zinc borate glasses using the numerical approach. The idea is fine, but some important parts are missing or need revision in both research design and writing. Some examples are listed below, but the authors are recommended to go through the whole paper again to address any other similar issues:

  1. For the simulation in 2.2, did you use any experiments to characterize glass properties as simulation input, or use theoretical values?

Response: Dear Reviewer, many thanks for your questions and effort to improve the quality of paper. In the simulation input, we need to know elemental mass fractions (%wt.) and densities. Thus, the aforementioned properties were obtained from the reference paper and highlighted in Table 1 of our paper. The required part was highlighted by YELLOW in the main text. The remaining simulation details are general issues such as used tools and locations of those tools. This information was also provided in Figure 2.

  1. Line 141 to 156 is similar to the introduction, you may consider move this part upward and combine with the introduction. Actually, section 3 should start with simulation results directly.

Response: Dear Reviewer, the part between the 141-156 has been moved to introduction. Accordingly, the related part in the results and discussion has been revised.

  1. You mentioned in line 168 to 169 that difference in LAC values are higher in the high energy region, but I do not see that clearly from figure 3. You should use same y-scales in the two zoom-in plots for a reasonable comparison.

Response: Dear Reviewer, in this case that not possible to have the same scale. due to the big variation between the first and last sample at the highest energy and between the first and last sample at the lowest energy, thus range is not similar. So, if we use the same scale it will be the visualization problem.

  1. Difference in figure 4 and 7 are not clear. Please add zoom-in plots like you did for other figures.

Response: Dear Reviewer, the requested zooms haven added to figure 4 and 7

  1. How did you measure the number of particles mentioned in line 236? Experimentally or theoretically? Not clear here.

Response: Dear Reviewer, the sentence was revised as “The average photon flux was measured in the detection field by using F4 tally mesh (see Figure 2).” We measured this quantity by using the F4 Tally mesh, which is a detection command that provides the average photon flux in a point or cell. The detailed properties are available in the MCNPX guide, which can also be found in the references.

  1. It is suggested to put tables right after the paragraphs that first mention them, so the table and content connection will be more clear.

Response: Dear Reviewer, the tables have been put right after the paragraphs that first mention them

  1. In section 2.1 you show a figure of real materials. Did you conduct any experiments for material property characterization or simulation results validation.

Response: Dear Reviewer, the initial study was done by our one co-author G.Kilic. In the previous study, the material characterization was done. Accordingly, we evaluate the radiation attenuation properties of those glass samples. This study contains only the comprehensive simulation results.

  1. As far as I can see, the simulation procedure is quite standard and the modeling input is obtained using simple theoretical calculation. As a result, the conclusion you get (lower density leads to lower radiation shielding) is quite trivial. Can you add some more content, such as theory, a new modeling method or experiments to provide more in-depth analysis for this glass radiation shielding phenomenon?

Reviewer 2 Report

H.O. Tekin et al. Materials 2020

Introduction

1. The authors present a very concise literature review and justifications on the use of glass materials within radiation shielding. This reviewer recommends having a look into the recent literature of promising radiation shielding materials, such as (not limited):

https://doi.org/10.1016/j.nucengdes.2016.11.009

It is also advised to compare the results obtained in the paper (parameters such as MAC, LAC, Neff, T1/2 etc) with glasses with conventional materials such as concretes.

Materials and Methods

1. The text in-between lines 83-94 is more literature review and introduction than materials and methods;

2. Lines 94-95, confusing and unclear phrasing;

Results and discussion

1. Line 225-226: Caption of figure 6 "men free path (...)";

2. Figure 9 has several different sample codes within the legends of the figures, but this sample indexing is simply absent in the main paper.

3. Table 2 must present S.I. units for MAC. Some confusing between comma and dot for differentiate decimals;

4. Table 3: same as table 2, units required.

5. Suggesting moving tables to appendix.

6. Comparison with available literature data is definitively required in the discussion...

Conclusions

1. Where the authors present any "semiconducting property" within the paper?

2. It is suggested to completely rewrite the conclusions to focus on the results presented in the paper. 

Author Response

Reviewer 2

Comments and Suggestions for Authors

H.O. Tekin et al. Materials 2020

Introduction

  1. The authors present a very concise literature review and justifications on the use of glass materials within radiation shielding. This reviewer recommends having a look into the recent literature of promising radiation shielding materials, such as (not limited):

https://doi.org/10.1016/j.nucengdes.2016.11.009

It is also advised to compare the results obtained in the paper (parameters such as MAC, LAC, Neff, T1/2 etc) with glasses with conventional materials such as concretes.

Response: Dear Reviewer, The recent literatures in addition https://doi.org/10.1016/j.nucengdes.2016.11.009 have been added to introduction section and comparison will be add to result and discussion section

Materials and Methods

  1. The text in-between lines 83-94 is more literature review and introduction than materials and methods;

Response: Revised. Dear Reviewer, the related part was moved to introduction.

  1. Lines 94-95, confusing and unclear phrasing;

Response: Revised.

Results and discussion

  1. Line 225-226: Caption of figure 6 "men free path (...)";

Response: Revised.

  1. Figure 9 has several different sample codes within the legends of the figures, but this sample indexing is simply absent in the main paper

Response: Revised.

  1. Table 2 must present S.I. units for MAC. Some confusing between comma and dot for differentiate decimals;

Response: Revised.

  1. Table 3: same as table 2, units required.

Response: Dear Reviewer, since the values are coefficients, there is no unit to add. However, Table 2 has been revised according to your valuable comments.

  1. Suggesting moving tables to appendix.

Response: Dear Reviewer, Since the coefficients are important part of EBF and EABF fitting parameters, we kindly request from you to keep these tables in the main file for readers and scientific community

  1. Comparison with available literature data is definitively required in the discussion...

Response: Agreed and Revised.

Conclusions

  1. Where the authors present any "semiconducting property" within the paper?

Response: Dear Reviewer, we tried to mention about recent literature works to highlight the importance of this type of glass. However, to avoid from confusion, we have removed this sentence.

  1. It is suggested to completely rewrite the conclusions to focus on the results presented in the paper. 

Response: Revised.

Round 2

Reviewer 2 Report

The authors answered my questions.
Please pay attention I did not see figures 11 and 12 in the main revised manuscript.